

# Prevalence of malocclusion and occlusal traits in the early mixed dentition in Shanghai, China

Xin Yu[1,2,*], Hao Zhang[2,3,*], Liangyan Sun[2,4], Jie Pan[2,4], Yuehua Liu[2,4] and Li Chen[2,4,5]

[1] Department of Pediatric Dentistry, Shanghai Stomatological Hospital, Fudan University, Shanghai, China
[2] Oral Biomedical Engineering Laboratory, Shanghai Stomatological Hospital, Fudan University, Shanghai, China
[3] Department of Preventive Dentistry, Shanghai Stomatological Hospital, Fudan University, Shanghai, China
[4] Department of Orthodontics, Shanghai Stomatological Hospital, Fudan University, Shanghai, China
[5] Minhang Preventive Dental Clinic, Shanghai, China
[*] These authors contributed equally to this work.

## ABSTRACT

**Background**. Epidemiological data on malocclusion among Chinese children are scant. The aim of this study was to provide detailed information on the prevalence of malocclusion in early mixed dentition children in Shanghai, China.

**Methods**. A cross-sectional survey was conducted from September 2016 to April 2017, and 2,810 children aged 7- to 9- years were selected from 10 primary schools by cluster random sampling. Several occlusal parameters, including Angle molar relationship, overjet, overbite, open bite, anterior and posterior crossbite, midline displacement, scissors bite, and teeth crowding and spacing, were clinically registered by five calibrated orthodontic dentists.

**Results**. We found that 79.4% children presented one or more occlusal anomalies. Angle Class I, Class II and Class III molar relationship were recorded in 42.3%, 50.9% and 5.9% of the sample, respectively. The proportion of Class III increased from 5.0% at age 7 to 7.8% at age 9. In the sagittal plane, increased overjet >3 mm was observed in 40.8% subjects, while the prevalence of severe overjet (>8 mm), anterior edge-to-edge (zero overjet) and anterior crossbite were 5.2%, 8.1% and 10.5%, respectively. Vertically, deep overbite >2/3 overlap was found in 6.2% of the children and open bite in 4.3%. Boys exhibited a higher rate of overbite than girls. For the transversal occlusal anomalies, 36.1% of the children had a midline displacement, which was followed by posterior crossbite (2.6%) and scissors bite (1.0%). Teeth space discrepancies were also common anomalies and anterior crowding (>2 mm) affecting 28.4% of the children, while anterior spacing (>4 mm) affecting 9.5%. Girls showed a higher prevalence of anterior crowding and a lower frequency of teeth spacing than boys.

**Conclusions**. Our study demonstrated that malocclusion is prevalent among children in the early mixed dentition, and more health resources should be warranted to meet the challenge of prevention or early intervention of malocclusion.

Corresponding author
Li Chen, lichen_kq@fudan.edu.cn

## INTRODUCTION

Malocclusion is one of the most common oral disorders among children, and it affects not only the oral masticatory function but also the craniofacial development and facial appearance. Children with certain malocclusion traits appear to have more problems related to psychology and social interactions, and even their quality of life suffers when they reach adulthood (*Martins-Junior, Marques & Ramos-Jorge, 2012*; *Nguyen et al., 1999*; *Stenvik, Espeland & Berg, 2011*). For this reason, malocclusion is regarded as an emerging public health issue.

The mixed dentition is an important developmental stage to the undisturbed occlusal relationship. The eruption of the first permanent molar plays a critical role in maintaining the interarch space and the sagittal occlusal relationship. Several longitudinal observations have revealed that a substantial number of malocclusions occur during this period (*Dimberg et al., 2015*; *Dimberg et al., 2013*; *Gois et al., 2012*), and the accumulated evidence has indicated that early intervention starting from the mixed dentition would benefit the youngsters with Class III malocclusion, crossbite, crowding and posterior crossbite (*Gianelly, 2002*; *Keski-Nisula et al., 2008*; *Lippold et al., 2013*; *Mitani, 2002*).

Epidemiological information is essential for developing strategies and plans to promote oral health. In China, national or local surveys on dental caries and periodontitis have been carried out regularly (*Li & Wang, 2014*; *Zhou et al., 2018*). However, there is still insufficient information on the prevalence of malocclusions. Recently, we made an effort to investigate the malocclusion status of Shanghai preschool children and an extraordinarily high prevalence, 83.9%, was found (*Zhou et al., 2017*). In the current study, another cross-sectional survey was carried out to assess the prevalence of malocclusion and the distribution of occlusal traits among school children at the stage of early mixed dentition in Shanghai.

## MATERIALS AND METHODS

### Study sample

For the period of September 2016 to April 2017, a cluster random sampling was applied in this study. In brief, we chose five administrative districts in Shanghai city: three of them in the urban area (Hongkou, Putuo, and Jing'an districts) and two in the suburbs (Pudong and Minhang districts). Then, two primary schools in each district were randomly selected, and the students with the following characteristics were identified as candidates of this survey: (1) aged 7–9 years; (2) without a history of orthodontic treatment; (3) without craniofacial diseases; and (4) consensual participation of the children and their parents. In all, 2,810 children, including 1,479 boys and 1,331 girls, were recruited.

The protocol of this study was approved by the Ethics Committee of Shanghai Stomatological Hospital, Fudan University (Approval Number: 2015-0012). Written informed consent was signed by the parents of all the children who participated in the survey.

**Table 1** Definition of occlusal traits along with the criteria of malocclusion.

| Occlusal traits | Definition | Malocclusion |
|---|---|---|
| **1. Sagittal anomalies** | | |
| 1.1 First permanent molars | Class I, the mesiobuccal cusp of the maxillary first permanent molar occludes with the mesiobuccal groove of the mandibular first permanent molar (normal relation), or up to or equal to 1/2 cusp width post-normal or pre-normal relation; Class II (distal), more than 1/2 cusp width post-normal relation; Class III (mesial), more than 1/2 cusp width pre-normal relation. | Class III |
| 1.2 Increased overjet | Distance of the most protruded maxillary incisor to the corresponding mandibular incisor: 0 mm, edge-to-edge (upper incisal edges touch lower edges when biting); >0 mm, ≤3 mm, normal; >3 mm, ≤5 mm, mild; >5 mm, ≤8 mm, moderate; >8 mm, severe | >3 mm |
| 1.3 Anterior crossbite | One or more of the maxillary incisors/canine occluded lingually to the mandibular incisors/canine. | Present |
| **2. Vertical anomalies** | | |
| 2.1 Deep overbite | Coverage of the mandibular incisors by most of the maxillary incisors: >0, ≤1/3, normal; >1/3, ≤1/2, mild; >1/2, ≤2/3, moderate; >2/3, severe | >2/3 |
| 2.2 Open bite | Negative vertically overlapping between the maxillary and the mandibular incisors: >0, ≤3 mm, mild; >3 mm, ≤5 mm, moderate; >5 mm, severe | >0 mm |
| **3. Transversal anomalies** | | |
| 3.1 Midline displacement | Mandibular midline deviated 2 mm or more to the maxillary midline | Present |
| 3.2 Posterior crossbite | One or more of the maxillary molars occluded lingually to the mandibular molars | Present |
| 3.3 Scissors bite | Maxillary molars occluded to the buccal surfaces of the corresponding mandibular molars, and/or mandibular molars occluded to the lingual surfaces the corresponding maxillary molars | Present |
| **4. Space discrepancies** | | |
| 4.1 Crowding (anterior, posterior; maxillary, mandibular) | >0 mm, ≤2 mm, mild; >2 mm, ≤4 mm, moderate; >4 mm, severe | >2 mm |
| 4.2 Anterior spacing (maxillary, mandibular) | >0 mm, ≤2 mm, mild; >2 mm, ≤4 mm, moderate; >4 mm, severe | >4 mm |

## Oral examination

The oral examination was carried out by five calibrated orthodontic dentists. The children were examined at schools, using portable lighting and disposable mouth mirrors. Sagittal molar relationships by Angle classification, degree of overjet and overbite, anterior and posterior crossbite, and teeth crowding and spacing were recorded (Table 1).

The children who presented one or more of the following indications were registered as malocclusion: Angle Class III, increased overjet (>3 mm), anterior crossbite, anterior edge-to-edge, deep overbite (>2/3 overlap), open bite, midline displacement, posterior crossbite, posterior edge-to-edge, scissors bite, anterior or posterior crowding (>2 mm), and anterior spacing (>4 mm).

**Table 2** Prevalence of malocclusion in 7–9-year-old children in Shanghai.

| | n | Normal occlusion | | Malocclusion | | P |
|---|---|---|---|---|---|---|
| | | n | % | n | % | |
| Age (years) | | | | | | 0.354[a] |
| 7 | 937 | 190 | 20.3 | 747 | 79.7 | |
| 8 | 1,217 | 241 | 19.8 | 976 | 80.2 | |
| 9 | 656 | 148 | 25.6 | 508 | 77.4 | |
| Gender | | | | | | 0.624[a] |
| Boys | 1,479 | 310 | 21.0 | 1,169 | 79.0 | |
| Girls | 1,331 | 269 | 20.2 | 1,062 | 79.8 | |
| Total | 2,810 | 579 | 20.6 | 2,231 | 79.4 | |

**Notes.**
[a] Chi-squared test.

## Reliability of examinations

Twenty subjects were evaluated by the five examiners independently of each other. One of the examiners was an orthodontist with more than fifteen years' clinic experience, and the other four examiners compared their results to the senior orthodontist's data respectively. Inter-examiner reliability was determined by calculating Cohen's kappa coefficient, and the values were >0.68.

## Statistical analysis

The rates of occlusal characteristics and malocclusion were reported by age and gender. The chi-squared test and Fisher's exact probability method were applied to determine the statistical associations between the independent variables and the malocclusion variable. Cohen's kappa value was used to measure the agreement among examiners. The data were input using the Epidata software and analyzed using SPSS Statistics 22 (IBM, Armonk, NY, USA). The level of significance was set at $p < .05$.

## RESULTS

The overall prevalence of malocclusion among school children aged 7–9 years in Shanghai was 79.4% (2231/2810), and only 20.6% of them had normal occlusion (Table 2). The boys had a very similar rate of malocclusion to that of the girls. No significant difference was observed between age groups ($p > .05$).

The distribution of the sagittal occlusal features among the children in Shanghai is shown in Table 3. The relationship of the first molars was classified according to the Angle classification; 42.3% children showed a Class I relationship, 50.9% children were Class II, and 5.9% were Class III. An increasing trend in the rate of Angle Class III with age was observed, from 5.0% at age 7 to 7.8% at age 9. The increased overjet was prevalent (40.8%), and most of the cases were mild or moderate, but 5.2% of the children were found to have a severe overjet. Approximately one-tenth of the children had an anterior crossbite.

Table 4 depicts the vertical and transversal occlusal anomalies. The probability of the deep overbite of the anterior teeth was 43.8% and that of severe overbite was 6.2%. Boys were more prone to deep overbite than girls ($p = .003$). The rate of open bite of anterior
**Table 3 Composition and prevalence of sagittal occlusal characteristic in 7–9-years-old children in Shanghai.**

| Sagittal occlusal characteristic | Age (years) | | | P | Sex | | P | Total | |
|---|---|---|---|---|---|---|---|---|---|
| | 7 | 8 | 9 | | Boys | Girls | | n | % |
| First permanent molar | | | | 0.017[c] | | | 0.361[d] | | |
| Normal (Class I) | 404 (43.1%) | 488 (40.1%) | 298 (45.4%) | | 647 (43.7%) | 543 (40.8%) | | 1,190 | 42.3 |
| Distal (Class II) | 474 (50.6%) | 650 (53.4%) | 306 (46.6%) | | 734 (49.6%) | 696 (52.3%) | | 1,430 | 50.9 |
| Mesial (Class III) | 47 (5.0%) | 68 (5.6%) | 51 (7.8%) | | 86 (5.8%) | 80 (6.0%) | | 166 | 5.9 |
| Mixed[a] | 3 (0.3%) | 4 (0.3%) | 0 (0.0%) | | 2 (0.1%) | 5 (0.4%) | | 7 | 0.2 |
| Lost/Not erupted[b] | 9 (1.0%) | 7 (0.6%) | 1 (0.2%) | | 10 (0.7%) | 7 (0.5%) | | 17 | 0.6 |
| Increased overjet | | | | 0.049[d] | | | 0.413[d] | | |
| Edge to edge | 80 (8.5%) | 97 (8.0%) | 52 (7.9%) | | 128 (8.7%) | 101 (7.6%) | | 229 | 8.1 |
| Normal (>0 mm, ≤3 mm) | 486 (51.9%) | 604 (49.6%) | 345 (52.6%) | | 751 (50.8%) | 684 (51.4%) | | 1,435 | 51.1 |
| Mild (>3 mm, ≤5 mm) | 238 (25.4%) | 275 (22.6%) | 148 (22.6%) | | 340 (23.0%) | 321 (24.1%) | | 661 | 23.5 |
| Moderate (>5 mm, ≤8 mm) | 99 (10.6%) | 161 (13.2%) | 80 (12.2%) | | 190 (12.8%) | 150 (11.3%) | | 340 | 12.1 |
| Severe (>8 mm) | 34 (3.6%) | 80 (6.6%) | 31 (4.7%) | | 70 (4.7%) | 75 (5.6%) | | 145 | 5.2 |
| Anterior crossbite | | | | 0.894[d] | | | 0.876[d] | | |
| Absent | 836 (89.2%) | 1,093 (89.8%) | 586 (89.3%) | | 1,325 (89.6%) | 1,190 (89.4%) | | 2,515 | 89.5 |
| Present | 101 (10.8%) | 124 (10.2%) | 70 (10.7%) | | 154 (10.4%) | 141 (10.6%) | | 295 | 10.5 |

**Notes.**
[a] Child with Class II first molar relation on one side and Class III on the other side.
[b] One or more first molars were missing or did not fully erupt.
[c] Fisher's exact test.
[d] Chi-squared test.

teeth was 4.24%; it decreased with age, from 4.9% at age 7 to 2.7% at age 9. With respect to transversal anomalies, 36.1% of the children were found to have a midline displacement, and 2.6% had posterior crossbite. The prevalence of a scissors bite was relatively low (0.9%), but it increased with age.

Teeth crowding and spacing were prevalent among the children (Table 5). The prevalence of anterior crowding of >2 mm of the maxillary or mandibular teeth was 13.3% and 22.5%, respectively. In all, 28.4% of the children presented anterior crowding. Posterior crowding was less common, and only 0.2% of the subjects were found to have maxillary posterior crowding of >2 mm, and 1.0% was mandibular posterior. The rate of anterior spacing of >4 mm of the maxillary teeth was 8.0%, and that of the mandibular teeth was 3.0%. An increasing trend with age was observed for the rate of crowding of the upper anterior teeth,

**Table 4  Composition and prevalence of vertical and transverse anomalies in 7–9-years-old children in Shanghai.**

| | Age (years) | | | P | Sex | | P | Total | |
|---|---|---|---|---|---|---|---|---|---|
| | 7 | 8 | 9 | | Boys | Girls | | n | % |
| Deep overbite | | | | 0.127[a] | | | 0.003[a] | | |
| None | 94 (10.0%) | 133 (10.9%) | 60 (9.1%) | | 158 (10.7%) | 129 (9.7%) | | 287 | 10.2 |
| Normal (>0, ≤1/3) | 451 (48.1%) | 535 (44.0%) | 307 (46.8%) | | 629 (42.5%) | 664 (49.9%) | | 1,293 | 46.0 |
| Mild (>1/3, ≤1/2) | 216 (23.1%) | 294 (24.2%) | 137 (20.9%) | | 366 (24.7%) | 281 (21.1%) | | 647 | 23.0 |
| Moderate (>1/2, ≤2/3) | 117 (12.5%) | 178 (14.6%) | 115 (17.5%) | | 226 (15.3%) | 184 (13.8%) | | 410 | 14.6 |
| Severe (>2/3) | 59 (6.3%) | 77 (6.3%) | 37 (5.6%) | | 100 (6.8%) | 73 (5.5%) | | 173 | 6.2 |
| Open bite | | | | 0.225[b] | | | 0.222[b] | | |
| None | 890 (95.0%) | 1,162 (95.5%) | 638 (97.3%) | | 1,420 (96.0%) | 1,270 (95.4%) | | 2,690 | 95.7 |
| Mild (>0, ≤3 mm) | 41 (4.4%) | 52 (4.3%) | 17 (2.6%) | | 56 (3.8%) | 54 (4.1%) | | 110 | 3.9 |
| Moderate (>3, ≤5 mm) | 5 (0.5%) | 3 (0.2%) | 1 (0.2%) | | 2 (0.1%) | 7 (0.5%) | | 9 | 0.3 |
| Severe (>5 mm) | 1 (0.1%) | 0 (0.0%) | 0 (0.0%) | | 1 (0.1%) | 0 (0.0%) | | 1 | 0.04 |
| Midline displacement | 326 (34.8%) | 469 (38.5%) | 219 (33.4%) | 0.052[a] | 544 (36.8%) | 470 (35.3%) | 0.418[a] | 1,014 | 36.1 |
| Posterior crossbite | 22 (2.3%) | 29 (2.4%) | 23 (3.5%) | 0.280[a] | 41 (2.8%) | 33 (2.5%) | 0.628[a] | 74 | 2.6 |
| Scissors bite | 2 (0.2%) | 11 (0.9%) | 12 (1.8%) | 0.003[b] | 12 (0.8%) | 13 (1.0%) | 0.641[a] | 25 | 0.9 |

**Notes.**
[a] Chi-squared test.
[b] Fisher's exact test.

and the boys' probability of anterior crowding, either of the maxillary or of the mandibular teeth, was lower than that of the girls' ($p < .001$).

## DISCUSSION

The prevalence of malocclusion in different populations ranges from 21% to 90% (*Grippaudo et al., 2013*; *Perillo et al., 2010*; *Perinetti et al., 2008*; *Shalish et al., 2013*; *Thilander et al., 2001*), and this huge variation may largely be attributed to the discrepancies in the definitions of malocclusion and the methodologies applied. We found that the prevalence of malocclusion in early mixed dentition in Shanghai was as high as 79.4%, which was considerably higher than the rate of 71.2% among children with mixed dentition in a national survey in 2000 (*Fu et al., 2002*). This result was similar to the rate of malocclusion in the deciduous dentition, i.e., 83.9%, in the Shanghai area (*Zhou et al., 2017*). Our findings confirmed that malocclusion was one of the most common health problems in children and adolescents.

**Table 5  Composition and prevalence of space discrepancies in 7–9-years-old children in Shanghai.**

| | Age (years) | | | P | Sex | | P | Total | |
|---|---|---|---|---|---|---|---|---|---|
| | 7 | 8 | 9 | | Boys | Girls | | n | % |
| Maxillary anterior crowding | | | | <0.001[a] | | | <0.001[a] | | |
| None | 631 (67.3%) | 727 (59.7%) | 366 (55.8%) | | 973 (65.8%) | 751 (56.4%) | | 1,724 | 61.4 |
| >0, ≤2 mm | 201 (21.5%) | 311 (25.6%) | 199 (30.3%) | | 341 (23.1%) | 370 (27.8%) | | 711 | 25.3 |
| >2, ≤4 mm | 92 (9.8%) | 144 (11.8%) | 68 (10.4%) | | 127 (8.6%) | 177 (13.3%) | | 304 | 10.8 |
| >4 mm | 13 (1.4%) | 35 (2.9%) | 23 (3.5%) | | 38 (2.6%) | 33 (2.5%) | | 71 | 2.5 |
| Maxillary posterior crowding | | | | 0.011[b] | | | 0.364[b] | | |
| None | 920 (98.2%) | 1,193 (98.0%) | 655 (99.8%) | | 1,462 (98.9%) | 1,306 (98.1%) | | 2,768 | 98.5 |
| >0, ≤2 mm | 14 (1.5%) | 21 (1.7%) | 1 (0.2%) | | 14 (0.9%) | 22 (1.7%) | | 36 | 1.3 |
| >2, ≤4 mm | 2 (0.2%) | 2 (0.2%) | 0 (0.0%) | | 2 (0.1%) | 2 (0.2%) | | 4 | 0.1 |
| >4 mm | 1 (0.1%) | 1 (0.1%) | 0 (0.0%) | | 1 (0.1%) | 1 (0.1%) | | 2 | 0.1 |
| Mandibular anterior crowding | | | | 0.006[a] | | | <0.001[a] | | |
| None | 373 (39.8%) | 583 (47.9%) | 313 (47.7%) | | 736 (49.8%) | 533 (40.0%) | | 1,269 | 45.2 |
| >0, ≤2 mm | 330 (35.2%) | 367 (30.2%) | 212 (32.3%) | | 444 (30.0%) | 465 (34.9%) | | 909 | 32.3 |
| >2, ≤4 mm | 184 (19.6%) | 204 (16.8%) | 99 (15.1%) | | 227 (15.3%) | 260 (19.5%) | | 487 | 17.3 |
| >4 mm | 50 (5.3%) | 63 (5.2%) | 32 (4.9%) | | 72 (4.9%) | 73 (5.5%) | | 145 | 5.2 |
| Mandibular posterior crowding | | | | 0.141[b] | | | 0.991[a] | | |
| None | 917 (97.9%) | 1,191 (97.9%) | 648 (98.8%) | | 1,450 (98.0%) | 1,306 (98.1%) | | 2,756 | 98.1 |
| >0, ≤2 mm | 11 (1.2%) | 15 (1.2%) | 1 (0.2%) | | 15 (1.0%) | 12 (0.9%) | | 27 | 1.0 |
| >2, ≤4 mm | 7 (0.7%) | 9 (0.7%) | 7 (1.1%) | | 12 (0.8%) | 11 (0.8%) | | 23 | 0.8 |
| >4 mm | 2 (0.2%) | 2 (0.2%) | 0 (0) | | 2 (0.1%) | 2 (0.2%) | | 4 | 0.1 |
| Maxillary anterior spacing | | | | <0.001[a] | | | <0.001[a] | | |
| None | 487 (52.0%) | 760 (62.4%) | 434 (66.2%) | | 841 (56.9%) | 840 (63.1%) | | 1,681 | 59.8 |
| >0, ≤2 mm | 244 (26.0%) | 277 (22.8%) | 128 (19.5%) | | 356 (24.1%) | 293 (22.0%) | | 649 | 23.1 |
| >2, ≤4 mm | 115 (12.3%) | 102 (8.4%) | 39 (5.9%) | | 137 (9.3%) | 119 (8.9%) | | 256 | 9.1 |

**Table 5** (*continued*)

| | Age (years) | | | P | Sex | | P | Total | |
|---|---|---|---|---|---|---|---|---|---|
| | **7** | **8** | **9** | | **Boys** | **Girls** | | **n** | **%** |
| >4 mm | 91 (9.7%) | 78 (6.4%) | 55 (8.4%) | | 145 (9.8%) | 79 (5.9%) | | 224 | 8.0 |
| Mandibular anterior spacing | | | | 0.116[a] | | | 0.081[a] | | |
| None | 734 (78.3%) | 948 (77.9%) | 523 (79.7%) | | 1,136 (76.8%) | 1,069 (80.3%) | | 2,205 | 78.5 |
| >0, ≤2 mm | 129 (13.8%) | 183 (15.0%) | 82 (12.5%) | | 216 (14.6%) | 178 (13.4%) | | 394 | 14.0 |
| >2, ≤4 mm | 51 (5.4%) | 52 (4.3%) | 23 (3.5%) | | 75 (5.1%) | 51 (3.8%) | | 126 | 4.5 |
| >4 mm | 23 (2.5%) | 34 (2.8%) | 28 (4.3%) | | 52 (3.5%) | 33 (2.5%) | | 85 | 3.0 |

**Notes.**
[a] Chi-squared test.
[b] Fisher exact test.

Deep overbite (>1/3 overlap, prevalence 43.8%) and increased overjet (>3 mm, prevalence 40.8%) were the two most common types of occlusion abnormalities in Shanghai schoolchildren. The high rates of overbite and overjet were also reported in Nigeria (deep overbite: 31.7% and increased overjet: 44.6%) (*DaCosta et al., 2016*) and in China's western city Xi'an (deep overbite 37.6% and increased overjet 35.0%) (*Zhou et al., 2016*). Nevertheless, compared to the fact that 63.7% of the preschool children were found to have deep overbite in Shanghai (*Zhou et al., 2017*), it was less frequent in the age group considered in this study. This decline could be partly explained by the self-correction of deep overbite during dental development (*Dimberg et al., 2015*). The increased overjet (>3 mm) occurred more frequently in the early mixed dentition (43.8%) than in the primary stage (33.9%) (*Zhou et al., 2017*). However, in terms of severe increased overjet (>8 mm), the change was substantial: 0.9% for primary and 5.2% for mixed dentition. This change may increase the risk of oral trauma (*Nguyen et al., 1999*).

Anterior crowding and anterior crossbite were another two high-incidence malocclusions observed in this study. In contrast to deep overbite, crowding and crossbite are less likely to be self-corrected without any intervention or treatment. Anterior crowding of >2 mm was recorded in 28.4% of the subjects, considerably more frequent than the proportion in primary dentition in the city (*Zhou et al., 2017*). Moreover, the crowding problem might be worse in the permanent dentition stage, as the arch length decreased during the transition from the mixed to the permanent dentition (*Gianelly, 2002*). It was noteworthy that anterior crowding was more prevalent in the mandible (22.5%) than maxilla (13.3%) in Shanghai children, which was consistent in what was found among children in the early mixed dentition in Germany (*Tausche, Luck & Harzer, 2004*) and adolescents in the permanent dentition in Japan (*Komazaki et al., 2012*). However, in Iran and Turkey, adolescents had more crowding in the maxilla than mandible (*Borzabadi-Farahani, Borzabadi-Farahani & Eslamipour, 2009*; *Gelgor, Karaman & Ercan, 2007*).

The prevalence of anterior crossbite in Shanghai children was comparable to that in Israeli (9.5%) (*Shalish et al., 2013*), German (7.7%) (*Tausche, Luck & Harzer, 2004*), and

Iranian (8.4%) (*Borzabadi-Farahani, Borzabadi-Farahani & Eslamipour, 2009*) children. Nevertheless, only 2.6% of the children had a posterior crossbite, which was relatively less frequent comparing the rates in Canada (15%) (*Karaiskos et al., 2005*), Brazil (13.3%) (*Almeida et al., 2011*) and Israel (23.3%) (*Shalish et al., 2013*). Several studies have pointed out that Chinese adults have a higher prevalence of Angle Class III malocclusion than the other racial groups (*Lew, Foong & Loh, 1993*; *Soh, Sandham & Chan, 2005*; *Woon, 1988*); however, we found that this rate was acceptable in Shanghai children, even though it was slightly lower in children from other Asian countries (*Borzabadi-Farahani, Borzabadi-Farahani & Eslamipour, 2009*; *Komazaki et al., 2012*). We found about 50 percent of the children had an Angle Class II molar relationship, and the rate was much higher than those reported in Germany (28%) (*Tausche, Luck & Harzer, 2004*), Brazil (21.4%) (*Dias & Gleiser, 2009*), Israel (29.9%) (*Shalish et al., 2013*) and Sweden (28%) (*Dimberg et al., 2013*). A high prevalence of Angle Class II, namely 38.2%, was also reported among 12 to 15-year-old adolescents in Japan (*Komazaki et al., 2012*), and it seemed that East Asians were more prone to have Angle Class II.

Although there was no difference of overall prevalence of malocclusion between boys and girls, several sexual dimorphisms were identified in the current study. It seemed that boys were more likely to have an overbite than girls, and this finding was supported by previous studies in Germany, France, Turkey and Brazil (*Dias & Gleiser, 2009*; *Gelgor, Karaman & Ercan, 2007*; *Lux et al., 2009*; *Souames et al., 2006*). Nevertheless, anterior crowding was more prevalent among girls than boys, which was consistent with what was found in Japan and Colombia (*Komazaki et al., 2012*; *Thilander et al., 2001*). These dimorphisms might be explained by the differences in skeletal maturity and/or eruption of permanent teeth (*Lux et al., 2009*).

Despite the reported benefit of early intervention of malocclusion (*Dimberg et al., 2013*; *Keski-Nisula et al., 2008*; *Proffit, 2006*), the high prevalence of malocclusion did not mean that most children were subjected to orthodontic treatment. Since these children were in the "ugly duckling" stage, and they probably suffered transient malocclusions, and some of them, such as maxillary midline diastema, increased overjet, deep overbite, crowding and even Angle Class II molar relationship, might be spontaneously corrected (*Dimberg et al., 2015*; *Huang & Creath, 1995*; *Kapur et al., 2018*). On the other hand, treatment priorities may vary depending on the severity of malocclusions. Therefore, many investigators have considered the orthodontic treatment need indices such as the Index of Orthodontic Treatment Need (IOTN) in epidemiological studies (*DaCosta et al., 2016*; *Komazaki et al., 2012*; *Shalish et al., 2013*; *Steinmassl et al., 2017*; *Tausche, Luck & Harzer, 2004*; *Thilander et al., 2001*). Even though the assessment of the orthodontic treatment need was not the major aim of the current survey, we attempted to obtain a rough estimate of this need on the basis of the criteria of IOTN's Grade 4 and Grade 5 and found that 26.2% of the children exhibited one or more of the following conditions (Table S1): Angle Class III, increased overjet >8 mm, anterior crossbite, open bite >3 mm, posterior crossbite, scissors bite, and anterior or posterior crowding >4 mm. This rate was consistent with that in

the Germans (26.2%) (*Tausche, Luck & Harzer, 2004*), the Iranians (23%) (*Borzabadi-Farahani, Borzabadi-Farahani & Eslamipour, 2009*), and the Austrians (30.6%) (*Steinmassl et al., 2017*).

A strict cluster random sampling was conducted, and a good representation was obtained in this study. Since this survey was school-based, it was infeasible to obtain the treatment records from the children who had a history of orthodontic intervention, and we excluded them because we did not know their original occlusal traits which had already been changed. Although many investigators did so in previous epidemiological studies (*Komazaki et al., 2012*; *Lagana et al., 2013*; *Souames et al., 2006*; *Thilander et al., 2001*), it should be kept in mind that this exclusion may introduce some representativeness bias. However, to the best of our knowledge, very few children under the age of 10 years appeal to orthodontists for malocclusion in Shanghai, the effects of the exclusion may be limited. Actually, in the current study, no more than one percent of the subjects had received orthodontic treatment, which was less than the rate reported in French children (*Souames et al., 2006*).

A large number of young people meet the criteria for early orthodontic treatment, and this is a huge challenge for our health system. Besides early treatment, establishing effective policies to prevent the occurrence of malocclusion may be another choice. Multiple factors, including genetic, environmental, and social-behavioral factors, play a role in the development of malocclusion (*Grippaudo et al., 2016*; *Lagana et al., 2013*). Some feeding habits and oral habits are believed to be important causes of malocclusion, and sucking habits are associated with anterior open bite and posterior crossbite (*Agarwal et al., 2014*; *Boronat-Catala et al., 2017*; *Gungor, Taner & Kaygisiz, 2016*). Therefore, attention needs to be paid to malocclusion disorders, and early health education and behavior intervention may contribute to a reduction of the burden of malocclusion.

## CONCLUSIONS

Our cross-sectional study demonstrated that 79.4% of the children in the stage of mixed dentition had one or more malocclusion traits. For the prevention and intervention of malocclusion, substantial resources and efforts are warranted from orthodontists, health policy makers, communities, and, of course, families.

## ACKNOWLEDGEMENTS

The authors would like to express their sincere gratitude to all the workers of the five Preventive Dental Clinics in the Hongkou, Putuo, Jing'an, Pudong, and Minhang districts for supporting this study. We thank LetPub for its linguistic assistance during the preparation of this manuscript.

### Funding

This study was supported by Projects of Development and Application of Suitable Health Technology of Shanghai Hospital Development Center (SHDC) (No. SHDC12014226),

Three Year Action Plan for Strengthening Public Health System in Shanghai (No. GWIV-12), Projects of Shanghai Municipal Commission of Health and Family Planning (No. 2016ZB0102-01) and Project of Characteristic Medical Specialty in Minhang Shanghai (No. 2017MWTZ20). The funders had no role in study design, data collection and analysis, decision to publish, or preparation of the manuscript.

### Grant Disclosures

The following grant information was disclosed by the authors:
Projects of Development and Application of Suitable Health Technology of Shanghai Hospital Development Center (SHDC): SHDC12014226.
Strengthening Public Health System in Shanghai: GWIV-12.
Projects of Shanghai Municipal Commission of Health and Family Planning: 2016ZB0102-01.
Project of Characteristic Medical Specialty in Minhang Shanghai: 2017MWTZ20.

### Competing Interests

The authors declare there are no competing interests.

### Author Contributions

- Xin Yu performed the experiments, prepared figures and/or tables, authored or reviewed drafts of the paper, approved the final draft.
- Hao Zhang performed the experiments, analyzed the data, prepared figures and/or tables, authored or reviewed drafts of the paper, approved the final draft.
- Liangyan Sun and Jie Pan performed the experiments, approved the final draft.
- Yuehua Liu conceived and designed the experiments, authored or reviewed drafts of the paper, approved the final draft.
- Li Chen conceived and designed the experiments, performed the experiments, authored or reviewed drafts of the paper, approved the final draft.

### Human Ethics

The following information was supplied relating to ethical approvals (i.e., approving body and any reference numbers):

This study was approved by the Ethics Committee of Shanghai Stomatological Hospital, Fudan University (2015-0012).

### Data Availability

The raw data is available in the Supplemental Files.

### Supplemental Information

Supplemental information for this article can be found online at http://dx.doi.org/10.7717/peerj.6630#supplemental-information.

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
