# Peer review of "Prevalence of malocclusion and occlusal traits in the early mixed dentition in Shanghai, China"

_PeerJ, doi:10.7717/peerj.6630_

## Round 0.1 · original submission · Major Revisions

Reviewer 1 and 3 have made a number of useful suggestions that could improve your paper. I would like to add the following:
* the manuscript needs extensive editing by a professional academic copy-editing service. I have good experience with SFedit.net but you are free to choose any professional service you like.
* as reviewer 1 suggested have a look at the reporting in other papers on the same subject and revise accordingly.
* the intra-observer and interobserver agreement are not very well described. Please describe the procedure: how many observers did how many double measurements and in the statistical section describe the test.
* Please pay attention to the definitions of the variables. As pointed out by the referee the definitions are not clear. For example variable 3.4: buccal surfaces, which surfaces upper or lower molars?
* ref 4 is missing last names of two authors

Reviewer 1 ·

Basic reporting

there are many grammar and style issues, revise,

instead of 'maxillary crowing >2mm of anterior' revise to'anterior maxillary crowing >2mm'
deficient literature review, why you didnt use indices?

the title is misleading you need to change it to 'Prevalence of malocclusion in the early mixed dentition and selected severely deviated occlusal traits in Shanghai, China'

M&M, add subheadings for sample size calculation, study sample, the variables you measured, statistical analysis
results: you need to provide more information for the readers:

abstract needs revision, 'discussion' should change to' conclusion'

increased overjet (how did you define it?) (40.8%), anterior crossbite (10.5%), anterior edge to edge (confusing terminology, revise, how did you define it?) (8.1%), deep overbite >2/3 overlap (6.2%), class III (how did you define it?, what classification did you use, angle or British system(incisors)) (5.9%), open bite (4.3%), midline irregularity(what is this?) (36.1%), scissors or opposite scissors bite (very confusing, revise) (1.0%), posterior crossbite (2.0%), posterior edge to edge (0.6%), maxillary crowing >2mm of anterior (13.3%) and posterior teeth (0.2%), mandibular crowing >2mm of anterior (22.5%) and posterior teeth (1.0%), anterior spacing >4mm in maxilla (8.0%) and in mandible (3.0%).

what was the overall, prevalence in comparison to permanent detention(about 30%, Prog Orthod. 2011 Nov;12(2):132-42.;Eur J Paediatr Dent. 2009 Jun;10(2):69-74.)

Experimental design

fine

Validity of the findings

fine

Additional comments

revise the paper, look at similar papers and revise similarly

·

Basic reporting

1) Professional English needs to be improved, edit this paper by an expert in Scientific English language

2) Some references are too old, insert other epidemiological papers such as :
- Prevalence of malocclusions, oral habits and orthodontic treatment need in a 7- to 15-year-old schoolchildren population in Tirana.
Laganà G, Masucci C, Fabi F, Bollero P, Cozza P.
Prog Orthod. 2013 Jun 14;14:12. doi: 10.1186/2196-1042-14-12.
- Oral habits in a population of Albanian growing subjects.
Laganà G, Fabi F, Abazi Y, Beshiri Nastasi E, Vinjolli F, Cozza P.
Eur J Paediatr Dent. 2013 Dec;14(4):309-13.

3) Table 1 needs to be modified, remove the text, it is too much for a table

4) Seven tables are too many, you can represent more results in a single table

Experimental design

Clarify better the "consistency and reliability" of the five orthodontists

Validity of the findings

The results are important in the Chinese area, because of recent informations' lack
In Discussion section, Authors can discuss the results comparing with different countries

Reviewer 3 ·

Basic reporting

The aim of the paper to evaluate the prevalence of malocclusion in the Chinese population is important, however there are several shortcomings: 1/ the modification of the IOTN used is not explained; 2/ the calibration is not described; 3/ no effort was made to look at the pretreatment records of the "few" children who already had orthodontic treatment; 4/ the term "individual normal occlusion" was not explained; 5/ some of the significant results were not explained in the discussion, ie increase of overjet with age, increase in crowding in boys in comparison to girls etc .

Experimental design

1/ The method used is "modified IOTN". It is not clear what kind of modification was preformed and was it validated before it was used in this study?
2/ The calibration of the 5 dentists is not described.
3/ No effort was made to look at the pretreatment records of the "few" children who already had orthodontic treatment.
4/ The term "individual normal occlusion" guided the authors in their decision to treat or not to treat. However, this term in not explained in the text.
5/ Functional shift was not examined although it could possibly explain some of the results like "mixed" molar relationships.

Validity of the findings

1/ Lack of validation of the modified IOTN may compromise the results.

Additional comments

In order to make a balanced review of the literature, in the introduction section mention should be made also of malocclusions which do not have to be treated early (L Johnston, J Bowman and others).
There are significant findings whish are not discussed or explained.

---

## Round 0.2 · Minor Revisions

All reviewers hold the opinion that your paper has improved a lot after you made the revisions.

However, the reviewers suggested some additional minor revisions and I agree on all of them.

Please pay attention to the English grammar. As I suggested before professional copy editing will help. Please note, that PeerJ is not providing that service, it is the responsibility of the authors.

Reviewer 1 ·

Basic reporting

Good reading but needs edit proffing

Experimental design

Ok

Validity of the findings

Ok

Additional comments

Please revise the paper for grammar and style

·

Basic reporting

English has been corrected and improved
References now are correct
Tables are weel described

Experimental design

ok

Validity of the findings

ok

Additional comments

Authors' editing work on the paper was appreciated.

Reviewer 3 ·

Basic reporting

1/Define "slightly more frequent" for line 156-7. The overjet in the early mixed dentition was found in 43.8% while in the primary stage in 33.9%. Is this a slight difference? Possible causes for this difference like the "ugly duckling stage" are not discussed.
2/ In Table 1: in the sagittal malocclusion only class III is mentioned. What about class II?
3/ In Table 3 the number of the footnotes in the table do not fit the numbers at the bottom of the table.
4/ Table 4 refers to vertical problems but referes to midline displacement which is a transverse problem.

Experimental design

no comment

Validity of the findings

no comments

Additional comments

no comments

---

## Round 0.3 · Minor Revisions

I have checked your revisions and they seem to be OK. I have made some minor corrections in the text. Please check and accept if you agree. I cannot attach the Word files to this mail. I will ask the editorial office to send them to you.

---

## Round 0.4 · accepted · Accept

I checked the last revisions (revision 3) and I am happy to see that your manuscript is now ready for production.